# Why Rivers Disappear—Remote Sensing Analysis of Postmining Factors Using the Example of the Sztoła River, Poland

Michał Lupa * , Aleksandra Pełka, Mariusz Młynarczuk , Jakub Staszel  and Katarzyna Adamek

Department of Geoinformatics and Applied Computer Science, Faculty of Geology, Geophysics and Environmental Protection, AGH University of Krakow, Mickiewicza 30 Av., 30-059 Krakow, Poland; pelka@student.agh.edu.pl (A.P.); mlynar@agh.edu.pl (M.M.); jstaszel@agh.edu.pl (J.S.); kadamek@agh.edu.pl (K.A.)
* Correspondence: mlupa@agh.edu.pl

**Abstract:** The impact of mining effects is undoubtedly an important research topic, especially in the case of assessing the effects of postmining factors. This study examines the drought in the Olkusz region using satellite imagery (Sentinel-2) and remote sensing indices. The analysis reveals that the region experienced multiple types of drought, including hydrogeological drought due to groundwater level lowering caused by mining activities, agricultural drought resulting from insufficient soil moisture, hydrological drought characterized by reduced water flow in rivers, and meteorological drought linked to decreased precipitation and high temperatures. This study demonstrates the usefulness of optical imaging and remote sensing indices in monitoring and assessing drought conditions. The results indicate significant changes in vegetation health and water content, as well as alterations to the natural environment within the region. This research highlights the importance of considering both human-induced and natural factors when evaluating drought phenomena. Continued monitoring and expansion of the study area would provide valuable insights into the long-term effects of weather conditions and the broader impacts on the ecosystem.

**Keywords:** remote sensing; drought; postmining

## 1. Introduction

Monitoring and assessing drought conditions have become increasingly crucial in the context of climate change and anthropogenic activities [1–3]. Remote sensing technologies offer a powerful tool for these evaluations, allowing for the monitoring of large and inaccessible areas and providing quantitative data for analyses [4–6]. Hydrological and hydrogeological drought occurring within mining areas is a commonly encountered and hazardous phenomenon [7,8]. The exploitation of deposits conducted in underground mines involves dewatering, which results in changes in water conditions [9,10]. This, in turn, can have an impact on increases in surface water pollution, as well as the drainage of wetlands and marshy areas [11,12].

Gerwin [13] drew attention to the decrease in groundwater levels in the western brown coal mining district of the Rhineland, the largest brown coal mining area in Europe, which has affected an approximately 4400 km$^2$ area [14,15]. Similar issues have arisen in other mining regions in Germany, such as in the Lusatia region [16]. In this area, as mines are being progressively closed, the water level continues to decline, raising concerns about maintaining the minimum water level in the Sprewa River.

The above implies that the proper management of areas affected by mineral extraction, the rational use of groundwater resources, and ensuring necessary environmental protection even after the cessation of mining activities play a crucial role in regions heavily impacted by human intervention in nature. Accurate and continuous monitoring plays a significant role in the proper reclamation of postmining areas. Modern imaging methods such as aerial scanning or satellite imagery are utilized for this purpose. This issue

is addressed, for example, by Sun et al. [17], who describe a method for quantitatively determining the impact of mining activities on groundwater level declines and vegetation using Sentinel-2 time series in monitoring iron mining areas in Liaoning Nanfen (LNMA), Sanheming (IMMA), and Sichuan Hongge (SCMA). Jawecki et al. [18] report the use of LiDAR data to estimate water retention in the region of the town of Strzelin in southern Poland, which includes approximately 80 inactive quarries and around 270 sand mines.

The evaluation and estimation of droughts using remote sensing data with limited access to ground observations are an important direction of research [1,19,20]. Remote sensing indices have been widely used for drought monitoring, providing an efficient way to assess large areas over time [21–23]. Each index is designed to measure specific variables like vegetation health, soil moisture, or water content and thus offers different insights into drought conditions.

The Normalized Difference Vegetation Index (NDVI) is a key tool for assessing vegetation health and is widely used in environmental monitoring [24–26]. NDVI values decrease when vegetation is stressed, often because of drought conditions [27]. In a study by Tucker et al. [28], the researchers expanded the NDVI datasets to be compatible with MODIS and SPOT data, enhancing their applicability in assessing vegetation health during drought events. Modified Soil-Adjusted Vegetation Index 2 (MSAVI2) was designed to improve upon earlier vegetation indices; MSAVI2 has proven useful in monitoring areas with sparse or mixed vegetation [29]. This index is sensitive to soil background, allowing for more accurate assessments of drought, particularly in regions with less dense vegetation. The Moisture Stress Index (MSI) was introduced to monitor vegetation leaf water content and has been used for assessing water stress in plants [30]. The index aids in the identification of plant water content, which is critical in evaluating drought conditions. The Normalized Difference Water Index (NDWI) was introduced as a tool to monitor liquid water in vegetation; NDWI is commonly used to observe hydrological drought by detecting changes in water bodies [31]. This index can effectively track reductions in water level or flow, making it invaluable for drought monitoring. The Normalized Multiband Drought Index (NMDI) provides a sensitive measure of moisture conditions, particularly useful in drought-affected regions [32]. This index has been employed in long-term studies to assess the impacts of drought over extended periods.

In the field of remote sensing applied to drought monitoring, these spectral indices have been tailored to capture various dimensions of drought phenomena. The Normalized Difference Water Index (NDWI) has two distinct variants, each serving specific purposes. For example, Islam and Sado [33] employed the first variant of NDWI to delineate open water features in Bangladesh and found it efficacious in tracking hydrological changes, thereby indirectly signaling drought conditions. Conversely, Xiao et al. [34] used the second NDWI variant to scrutinize water stress in rice paddy fields in South and Southeast Asia, demonstrating its utility in agricultural drought assessment. Similarly, the Normalized Difference Vegetation Index (NDVI) remains a prevalent tool in vegetation monitoring. Pettorelli et al. [35] used the NDVI to scrutinize ecological responses to environmental changes, concluding that diminishing NDVI values serve as robust indicators of drought-induced vegetation stress. Fensholt and Sandholt [36] also employed NDVI, along with other indices, to assess water stress in semiarid environments, corroborating its effectiveness in drought monitoring. Roerink et al. [37] used Modified Soil-Adjusted Vegetation Index 2 (MSAVI2) in the Sahel region to assess vegetation response to water availability, affirming its suitability for drought studies. The Normalized Multiband Drought Index (NMDI) has been adopted for long-term drought impact evaluations. For instance, Gu et al. [12] used the NMDI to examine a five-year drought cycle in the central Great Plains, establishing the index's sensitivity to moisture conditions. Similarly, Feng et al. [38] utilized the NMDI to corroborate Gravity Recovery and Climate Experiment (GRACE) data, concluding that it provides valuable insights into groundwater depletion, a key factor in prolonged droughts. Finally, the Moisture Stress Index (MSI) has been applied to detect plant water stress. Ceccato et al. [30] utilized the MSI to monitor vegetation leaf water content, identifying it

as an effective marker of drought-induced plant stress, while Peñuelas et al. [39] found it reliable for assessing water stress as an early sign of impending drought conditions.

A slightly different approach, using additional data, has been presented In many other works. In [40], satellite data and indices were utilized to assess drought in wetland areas, addressing the challenge of limited ground observations. The study employed the Vegetation Temperature Condition Index (VTCI), derived from satellite-based measurements of Land Surface Temperature (LST) and the Normalized Difference Vegetation Index (NDVI), to monitor drought patterns in Lake Chad's wetlands from 1999 to 2018. By integrating WorldView-3 imagery to evaluate the methodology, the approach achieved high accuracy (over 90%) in estimating spatially distributed VTCI values, highlighting its effectiveness in drought monitoring where conventional meteorological data are scarce. Another excellent example is presented by Bijaber [41], where the Composite Drought Indicator (CDI) was calculated for the purpose of monitoring drought in Morocco. The CDI utilizes the Normalized Difference Vegetation Index (NDVI) provided by the United States Geological Survey Earth Resources Observation and Science (USGS EROS) Center, generated by the Moderate Resolution Imaging Spectroradiometer (MODIS). It also incorporates Evapotranspiration (ET) anomalies derived from surface energy balance modeling, the Standardized Precipitation Index (SPI) using rainfall data from CHIRPSv2 (Climate Hazards Group InfraRed Precipitation with Stations), and Land Surface Temperature (LST) data from MODIS sensors. The indicator is used to create maps of the entire country, which are provided on a monthly basis, serving the assessment and monitoring of drought.

The present study focuses on the utilization of satellite imagery to describe the ecological catastrophe occurring in the postmining areas of the Olkusz district in southern Poland. It represents a critical case study because of its vulnerability to various forms of drought exacerbated by postmining activities. The cessation of mining operations and the subsequent stoppage of pumps have led to the disappearance of the Sztoła River. The authors want to show that, by using satellite methods, it is possible to study changes in post-industrial areas. This study aims to address these gaps by applying a multifaceted remote sensing approach to monitor and assess drought conditions in the region. Utilizing Sentinel-2 satellite imagery, the authors focus on a comprehensive set of remote sensing indices such as the NDVI, the MSAVI2, the Moisture Stress Index (MSI), the Normalized Difference Water Index (NDWI), and the Normalized Multiband Drought Index (NMDI) to gain a holistic view of the drought phenomenon in this unique setting.

## 2. Area of Interest

The study area, depicted in Figure 1, encompasses four municipalities within the Olkusz district, Bolesław (41 km$^2$), Bukowno (65 km$^2$), Klucze (119 km$^2$), and Olkusz (151 km$^2$) in the Lesser Poland Voivodeship, as well as one municipality in the Będzin district, Sławków (37 km$^2$), in the Silesian Voivodeship. The average annual temperature is 8 °C, and the growing season, defined as days with temperatures exceeding 5 °C, spans 200–210 days. Annual precipitation averages between 700 and 800 mm [42].

The topography of the study area exhibits variability, significantly influenced by human activities. Intensive mining operations have led to the formation of various anthropogenic features, predominantly excavations and spoil tips resulting from sand, zinc, and lead ore exploitation, along with quarries featuring limestone rocks [24]. The presence of limestone is linked to karst phenomena, manifested in monadnocks, caves, and sinkholes. Wood extraction for metallurgical purposes has exposed loose sands, contributing to the formation of the Błędowska Desert in the northern part of the study area [43]. There are four aquifers within the analyzed area, Quaternary, Jurassic, Triassic, and Paleozoic, with the Quaternary and Triassic being the most significant for this study. The Pomorzany Mine heavily exploited the Quaternary aquifer, resulting in substantial drainage and the formation of an extensive depression cone [44].

The limited river system in the Olkusz region is attributed to intense groundwater drainage and specific geological characteristics. The research area falls within the Biała

Przemsza River Basin, at the confluence of the Sztoła River, nourished by underground waters from the Baba River. Waters downstream from mining discharge points maintain a quality rating of very good or good. Despite pronounced mining and metallurgical activities related to zinc and lead ores, the study area holds significant natural value. Protected zones include areas within the Kraków Valleys Landscape Park and the Eagles' Nests Landscape Park, along with their buffer zone. The selection of the study area was influenced by the presence of a depression cone, and additionally, the western part was extended to incorporate the municipality of Sławków because of the Sztoła River's mouth location [45].

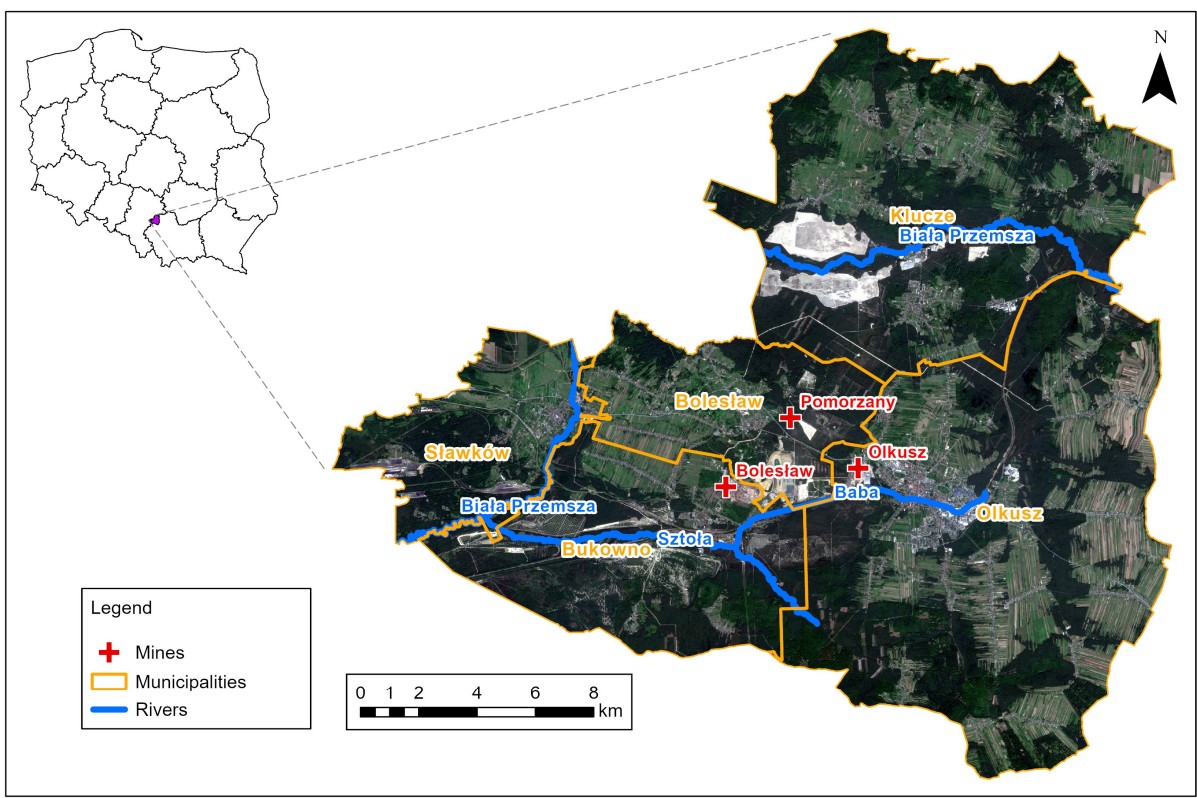

**Figure 1.** Area of interest. Sentinel-2 data acquired from AWS services.

## 3. Methods

The primary objective of the authors was to capture the environmental changes resulting from the discontinuation of water pumping from the Pomorzany Mine. These changes have frequently been the subject of media reports and interest from the local community. Moreover, given limited possibilities for in situ measurements, as well as the lack of research stations in the area and appropriate measuring instruments, the team decided to utilize a time series analysis of optical satellite images from the Sentinel-2 mission. The analyzed time series consisted of images obtained in May and August. This choice was dictated by two factors. The first was the phenomenon of meteorological drought, which occurs periodically in Poland. The second factor was the cloud cover level of the acquired images. Only in these two months was it possible to obtain data with a cloud cover level that allowed for analysis.

In the next step, six indices were calculated that could indicate problems with existing drought for images from the years 2017–2023. These indices were subsequently subjected to processing for the purpose of cloud masking and statistical calculations. The overall research scheme is presented in the figure below (Figure 2).

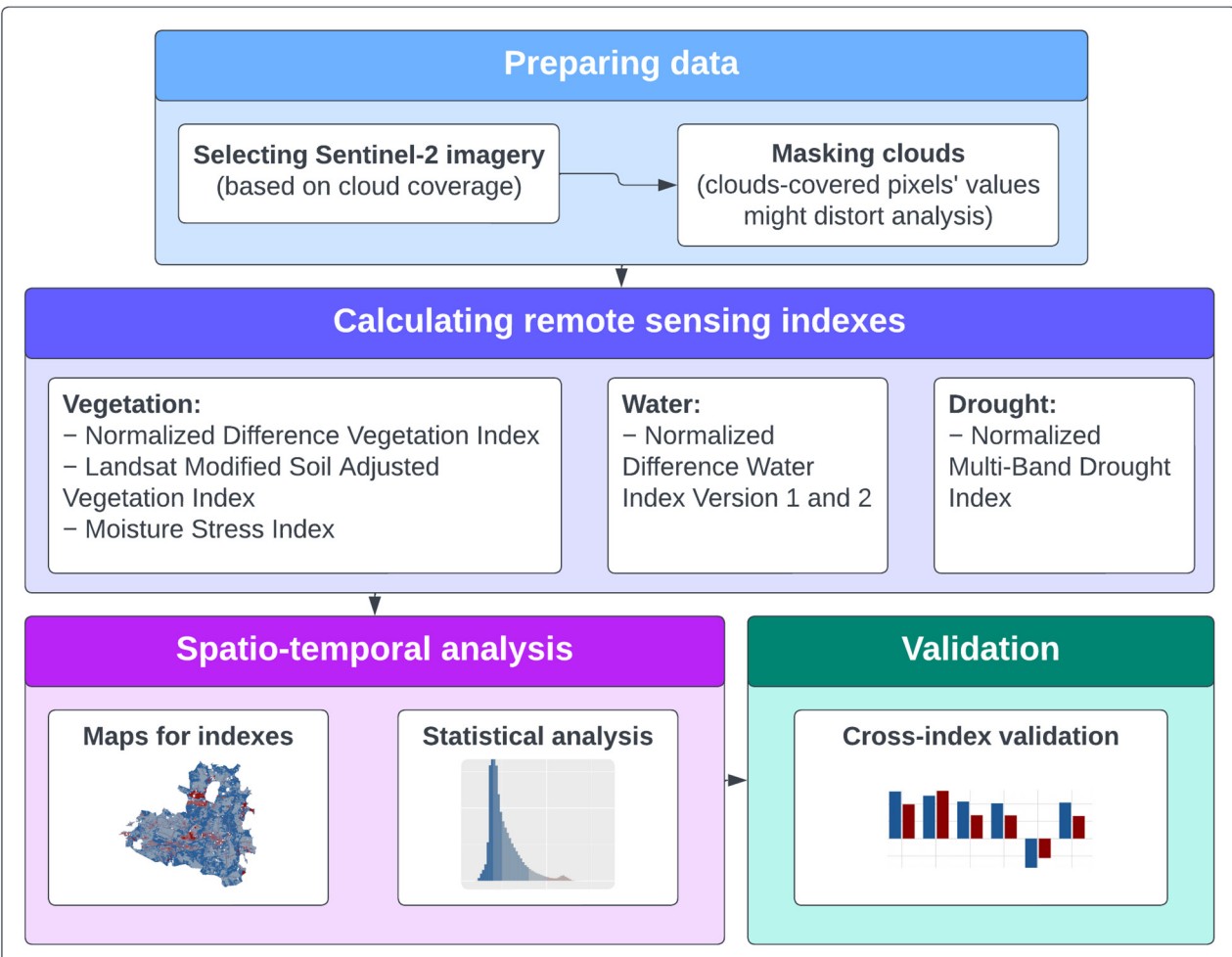

**Figure 2.** Methodological flow chart. A full set of comparative maps is presented in Supplementary Materials. Statistical analysis included computing average values and histograms. Cross-index validation checked if the changes in the environment, as evidenced by individual indices, were consistent.

### 3.1. Input Data

Data from the Sentinel-2 satellite mission for the years 2017–2023 for the months of May and September were used. The primary constraint imposed by the limited temporal scope of the analysis is the availability of satellite imagery from the Sentinel-2 mission. Concerning the monthly dimension, attention was directed toward changes in vegetation cover throughout the growing season, as well as the availability of imagery (varied levels of cloud cover throughout the year). The Level-2A product was used, which provides atmospherically corrected images. In a comparable growing season, representative images were selected for each year. In addition, weather data from one of the nearest meteorological stations—Kraków–Balice, located within a radius of 30 km from the study area—was used. For the analysis, the values of average monthly temperatures and monthly precipitation totals (for months falling into the growing season), obtained from the aforementioned station over the years 2017–2023, were used. In addition, the Land Cover Living Atlas (https://livingatlas.arcgis.com/landcover/, accessed on 10 September 2023) was used to assess land use. Based on the analysis of the available data, the impact of the change in land classification on the observations carried out in this study was excluded.

The analyses were carried out using the FME software and the R language. FME was used to determine 6 remote sensing indices (Table 1) and R to visualize the results.

**Table 1.** Formulas used to calculate remote sensing indices.

| Index | Sentinel-2 Bands | Formula | Source |
|---|---|---|---|
| NDWI (I variant) | B8A, B12 | $\frac{NIR-SWIR2}{NIR+SWIR2}$ | [46] |
| NDWI (II variant) | B03, B08 | $\frac{GREEN-NIR}{GREEN+NIR}$ | [31] |
| NDVI | B04, B08 | $(NIR-RED)/(NIR+RED)$ | [27] |
| MSAVI2 | B04, B08 | $\frac{2NIR+1-\sqrt{(2NIR+1)^2-8(NIR-RED)}}{2}$ | [29] |
| NMDI | B8A, B11, B12 | If NDVI $\geq$ 0.4: $\frac{NIR-(SWIR1-SWIR2)}{NIR+(SWIR1-SWIR2)}$ If NDVI $<$ 0.4: $0.9-\frac{NIR-(SWIR1-SWIR2)}{NIR+(SWIR1-SWIR2)}$ | [47] |
| MSI | B8A, B11 | $SWIR1/NIR$ | [30] |

- NDWI (I variant)

The Normalized Differential Water Index is used to monitor the water content in vegetation in drought areas. Low values of the presented index mean drought—the lower it is, the greater the intensity of this phenomenon, while values higher than 0.4 indicate the absence of drought [31].

- NDWI (II variant)

The Normalized Difference Water Index in variant II is used to detect open water reservoirs. Negative values of the NDWI indicator in the variant for monitoring water reservoirs stand for vegetation (the smaller it is, the better its condition) or areas affected by drought. A range from 0.0 to 0.2 indicates high humidity, while values higher than 0.2 correspond to water reservoirs [46].

- NDVI

The Normalized Difference Vegetation Index allows us to examine the condition of vegetation and assess its development phase. High NDVI values indicate greener and healthier vegetation. Values in a range from −1.0 to −0.1 correspond to areas of water, snow cover, or areas covered by clouds. A range from −0.1 to 0.1 indicates bare soil, rocks, and sand [48].

- MSAVI2

The Modified Soil Adjusted Vegetation Index makes it possible to assess the condition of vegetation in the early stages of development. Values in a range from −1.0 to 0.2 correspond to areas without vegetation, a range from 0.2 to 0.4 represents plants in the germination stage, and a range from 0.4 to 0.6 represents plants in the stage of leaf development. The highest values of MSAVI2 indicate dense vegetation [49].

- NMDI Soil

The Normalized Multiband Drought Index is used to monitor soil moisture and vegetation conditions. Unlike other indicators, the NMDI ranges from 0 to 1. Values less than 0.2 stand for very high drought and high fire risk. A range from 0.2 to 0.4 corresponds to drought, while 0.4–0.6 corresponds to moderately wet vegetation. Values higher than 0.6 indicate vegetation with very high water content [47].

- MSI

The Moisture Stress Index is used to estimate the water content of plants and assess the condition of vegetation. The MSI ranges from 0 to more than 3, with higher values indicating significant water stress, while values close to 0 characterize vegetation with high water content [50].

### 3.2. Methods Used to Evaluate Results

Apart from evaluating maps and statistics, a modification of average values was also used to facilitate the comparison and presentation of the results. The modification was performed separately for each of the indices. In each case, a vector of average index values was created from the consecutive satellite imagery used. Each vector was then normalized, and for some of the indices (MSAVI2, NDVI, NDWI (I variant)), the values were also inverted (values close to 0 are now close to 1). The average value for a given year is calculated as an average value for the exact index calculated for imagery in May and August. This operation allowed for the normalization of the values of all the indices into a form that was easier to interpret. Correlations between indices were assessed based on the Pearson coefficient calculated for flattened three-dimensional matrixes (where the third dimension was the index values for consecutive imagery used). This procedure allowed for the selection of a representative group of indices for the purpose of in-depth analysis but also enabled cross-index validation.

### 4. Results

Changes in the average values between the periods of 2017–2021 and 2022–2023 were observed for all the considered remote sensing indexes (Figure 3). In each case, these changes may indicate a deepening drought phenomenon and, among other things, worsening plant conditions, increased water stress, or reduced soil moisture. A higher risk of drought was observed for the year 2022, but the values for 2023 indicate no improvement in the condition of the environment (Figure 4).

**Figure 3.** Average values for indices calculated for the periods 2017–2021 and 2022–2023.

There were six remote sensing indices used in this research, but because of the high correlation between some of them (Table 2), only three were analyzed in depth: the MSI, NMDI Soil, and MSAVI2. The NDVI and the NDWI (II variant) are highly correlated (>0.9 | <−0.9) with MSAVI2. The NDWI (variant I) is highly correlated (<−0.9) with MSI and slightly less (>0.8) with MSAVI2 and the NDVI.

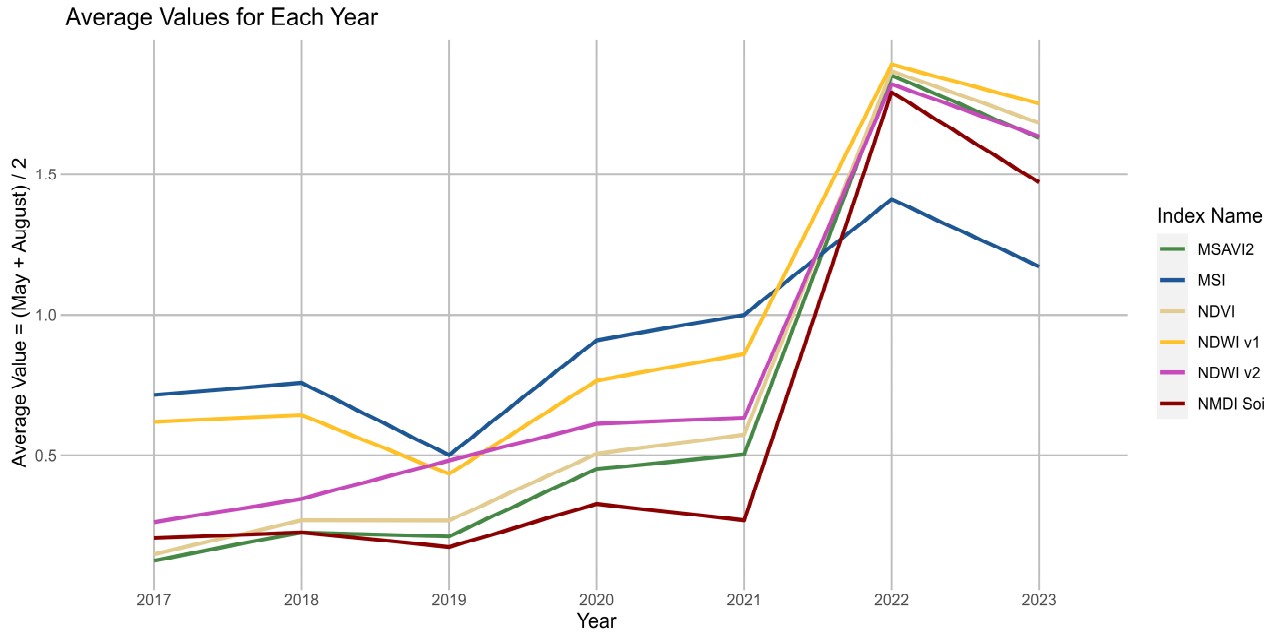

**Figure 4.** Average values for indices for each of the analyzed years (inverted values for MSAVI2, NDVI, and NDWI (I variant)).

**Table 2.** Pearson correlation values between indices.

| Index Name | MSAVI2 | MSI | NDVI | NDWI v1 | NDWI v2 | NMDI Soil |
|---|---|---|---|---|---|---|
| **MSAVI2** | - | −0.770 | 0.961 | 0.875 | −0.942 | 0.788 |
| **MSI** | −0.770 | - | −0.776 | −0.943 | 0.638 | −0.620 |
| **NDVI** | 0.961 | −0.776 | - | 0.891 | −0.935 | 0.785 |
| **NDWI v1** | 0.875 | −0.943 | 0.891 | - | −0.764 | 0.722 |
| **NDWI v2** | −0.942 | 0.638 | −0.935 | −0.764 | - | −0.737 |
| **NMDI Soil** | 0.788 | −0.620 | 0.785 | 0.722 | −0.737 | - |

*4.1. NMDI Soil—Soil Moisture Assessment*

The lower the average NMDI value, the drier the areas in the analyzed region. For the years 2017–2021, the wet and extremely wet classes (NMDI Soil $\geq$ 0.4) cover most of the analyzed area. In the two following years, the dominant classes are dry and extremely dry (NMDI Soil < 0.4). Particularly notable dry areas are found in the Olkusz area and in the valleys of the Biała Przemsza and Sztoła Rivers (Figures 5, 6 and S1). The average values of the NMDI for individual months (May and August) in the years 2017–2023 are presented in the table below (Table 3), and they confirm the mentioned change.

**Table 3.** Average values for NMDI Soil.

| Average Values | Month | 2017 | 2018 | 2019 | 2020 | 2021 | 2022 | 2023 |
|---|---|---|---|---|---|---|---|---|
| **NMDI Soil** | May | 0.539 | 0.542 | 0.538 | 0.486 | 0.479 | 0.333 | 0.340 |
| | August | 0.511 | 0.502 | 0.521 | 0.530 | 0.553 | 0.275 | 0.358 |

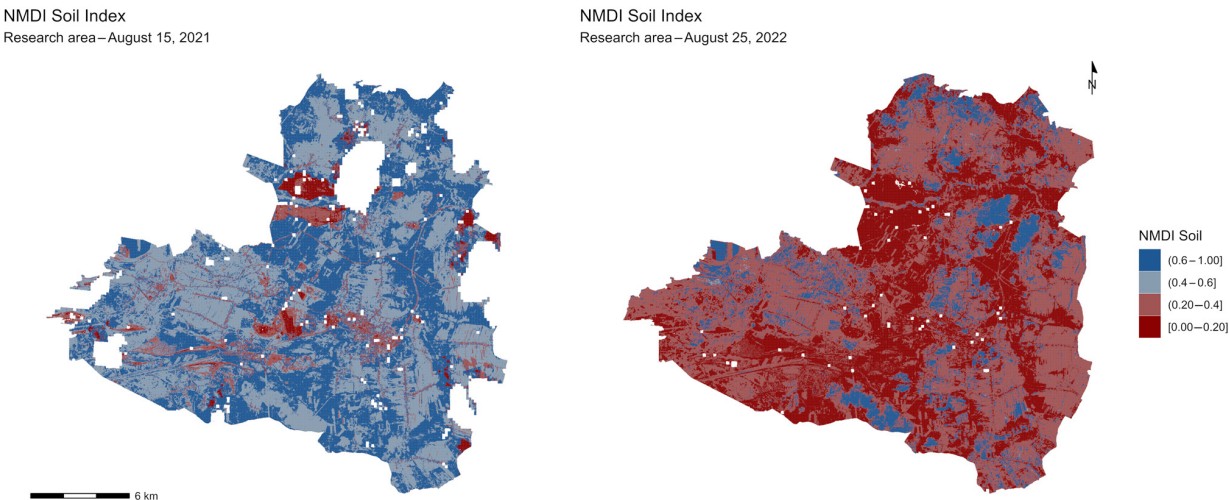

**Figure 5.** NMDI Soil maps comparing the values of this index between 15 August 2021 and 25 August 2022. Sentinel-2 data acquired from AWS services.

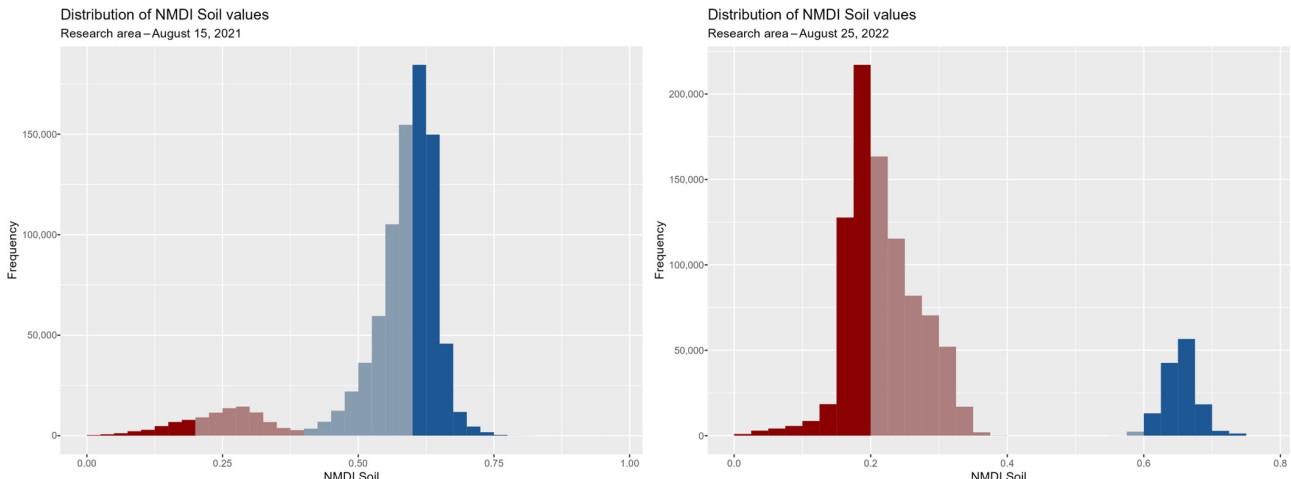

**Figure 6.** Histograms for NMDI Soil comparing the values of this index between 15 August 2021 and 25 August 2022.

### 4.2. MSAVI2—Vegetation Condition Assessment

The use of MSAVI2, which eliminates the impact of bare soil, allowed for the examination of the condition of even very small plants at various stages of development [49]. In the years 2017–2021, the study area was predominantly covered with classes >0.6 (dense vegetation) (Figures 7, 8 and S2). This was also confirmed by average values (Table 4). In 2022 and 2023 (compared with previous years), a weakening of the vegetation condition can be observed. The shares of classes 0.4–0.6 (leaf development) and 0.2–0.4 (germination phase) are higher (Figures 7 and S1), and the average values decrease toward 0.5 (Table 4). This may indicate that conditions occurred that limited growth and prevented plants from fully developing. In addition, a significant deterioration of the vegetation can be observed in the Sztoła River Valley compared with the Biała Przemsza River Valley (Figures 7 and S2).

**Table 4.** Average values for MSAVI2.

| Average Values | Month | 2017 | 2018 | 2019 | 2020 | 2021 | 2022 | 2023 |
|---|---|---|---|---|---|---|---|---|
| MSAVI2 | May | 0.718 | 0.719 | 0.688 | 0.614 | 0.598 | 0.499 | 0.505 |
| | August | 0.688 | 0.661 | 0.696 | 0.708 | 0.711 | 0.461 | 0.513 |

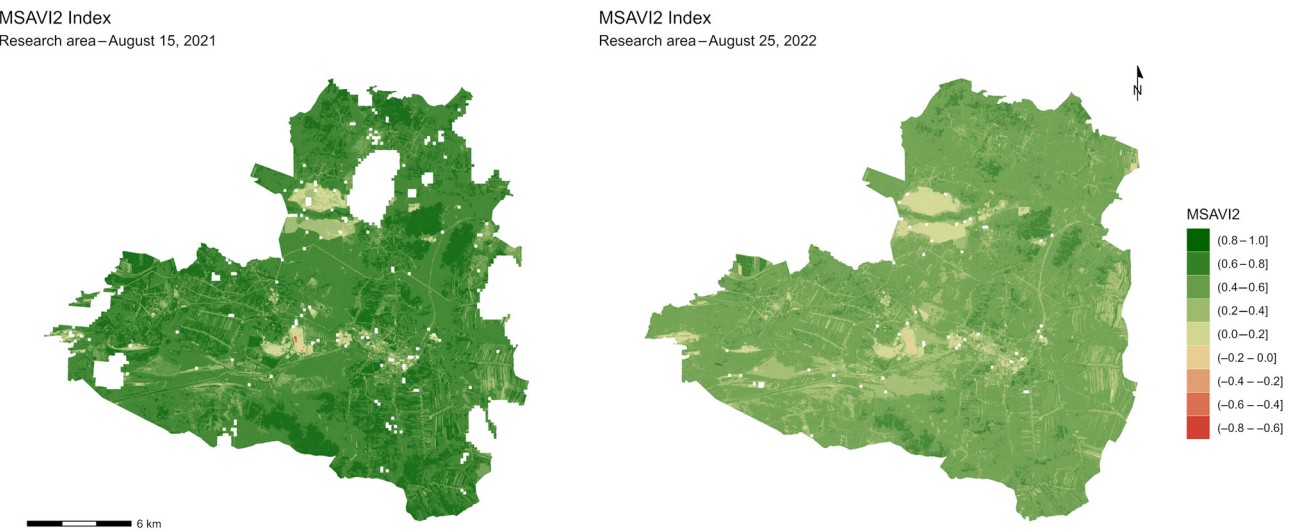

**Figure 7.** MSAVI maps comparing the values of this index between 15 August 2021, and 25 August 2022. Sentinel-2 data acquired from AWS services.

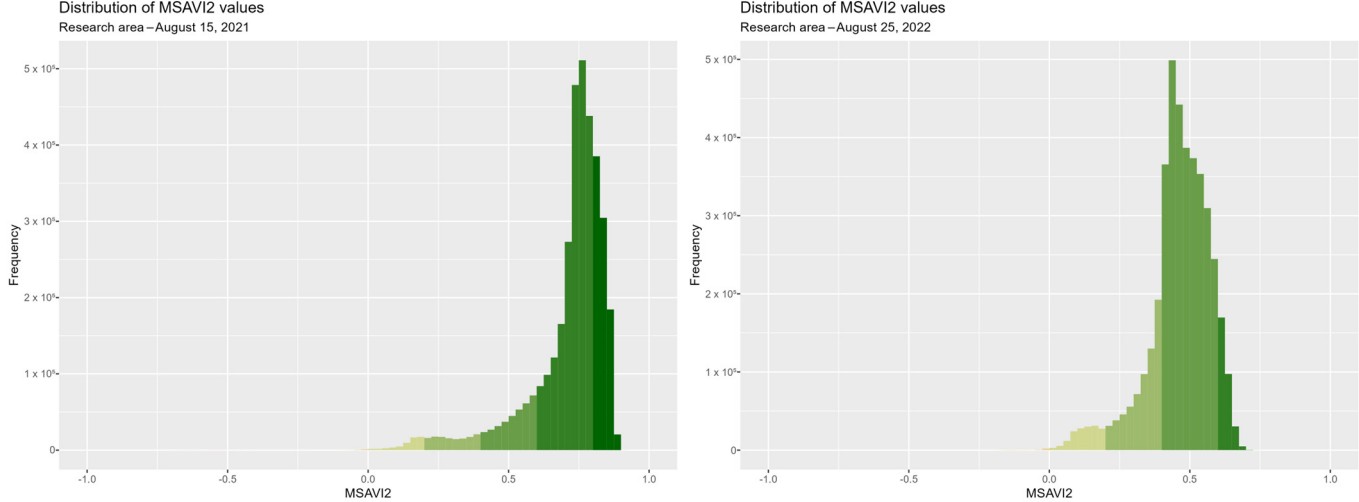

**Figure 8.** Histograms for MSAVI2 comparing the values of this index between 15 August 2021 and 25 August 2022.

### 4.3. MSI—Water Stress Assessment

A difference between the years 2017–2021 and 2022–2023 is noticeable for imagery from August (Table 5, Figures 9, 10 and S3). For the first five years, lower MSI values are noticeable for forested areas and river valleys, while higher values are observed for urban areas and cultivated fields. However, for the years 2022–2023, this distinction disappears, and the dominant class on the map becomes 0.5–0.8 (Figure 9). The changes described may indicate a significant increase in water stress (water availability for vegetation), which can have an impact on the deterioration of vegetation.

**Table 5.** Average values for MSI.

| Average Values | Month | 2017 | 2018 | 2019 | 2020 | 2021 | 2022 | 2023 |
|---|---|---|---|---|---|---|---|---|
| **MSI** | May | 0.597 | 0.600 | 0.580 | 0.702 | 0.784 | 0.717 | 0.666 |
| | August | 0.623 | 0.633 | 0.583 | 0.572 | 0.514 | 0.692 | 0.678 |

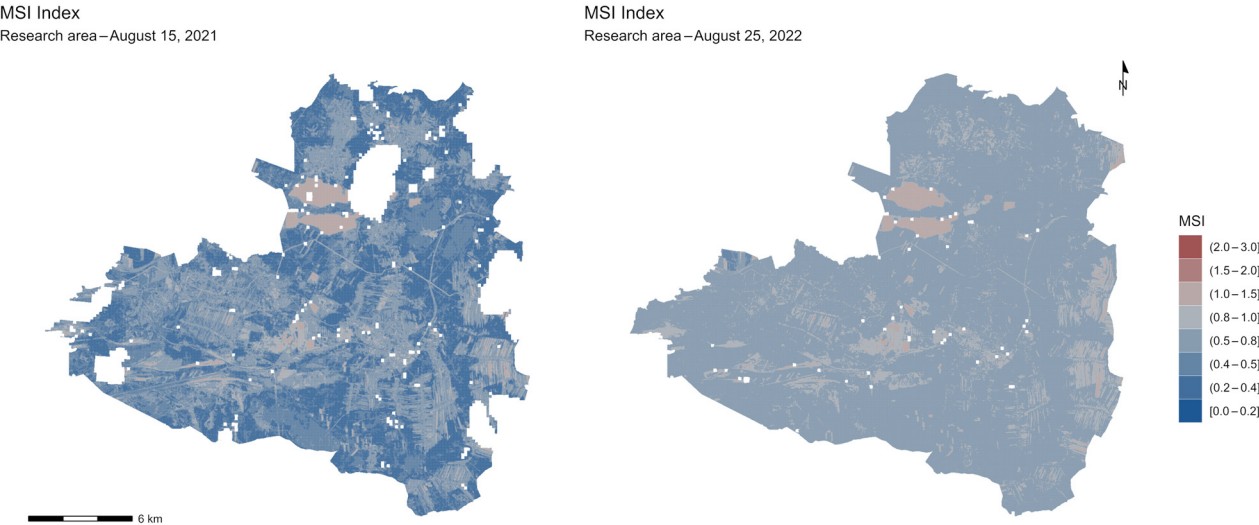

**Figure 9.** MSI maps comparing the values of this index between 15 August 2021 and 25 August 2022. Sentinel-2 data acquired from AWS services.

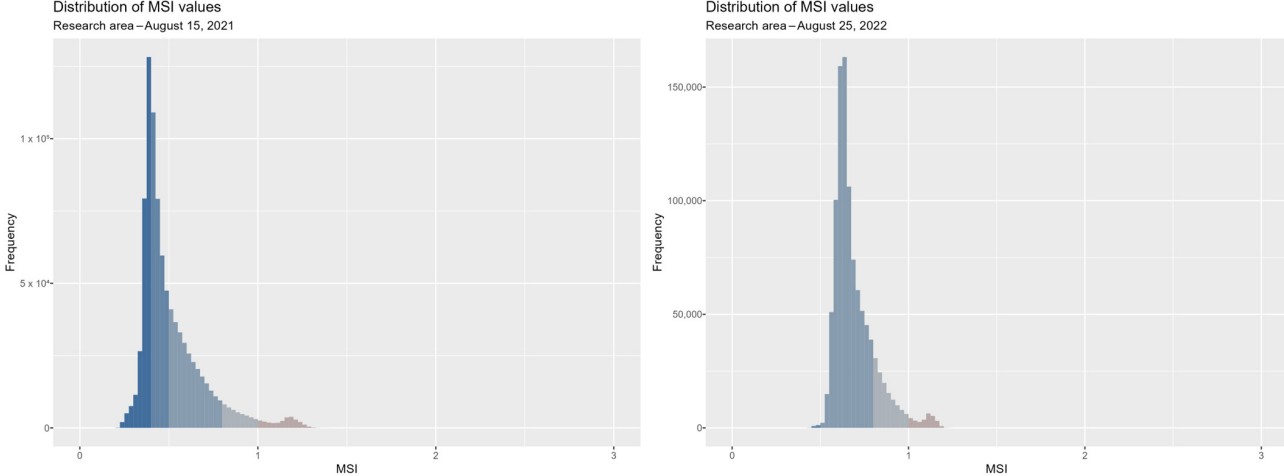

**Figure 10.** Histograms for MSI comparing the values of this index between 15 August 2021 and 25 August 2022.

### 4.4. Cross-Index Validation

Both indices related to assessing vegetation cover exhibit similar changes, with the average observed change for MSAVI2 being −27%, with −37% for the NDVI (Figure 3). Concerning water-related indices, the change in NDWI v1 was −34%, while for NDWI v2, it was +33%. The contrasting nature of these changes stems solely from the different interpretations of these indices. An increase in values for NDWI v2 in areas not covered by water does not indicate a higher water quantity but rather deteriorating vegetation quality. In the case of the MSI and the NMDI, indices with similar interpretations were not applied; however, regarding the analysis of temporal variability, they exhibit the same relationships as the other indices (Figure 3). The consistency of the changes indicated by all used indices is also reflected in the correlation values (Table 2).

## 5. Discussion

The application of remote sensing methods to monitor drought within the region turned out to be a very good solution because of the possibility of making a large-scale and long-term analysis of the state of the natural environment. The use of optical imaging, on the basis of which individual indicators were calculated, made it possible to estimate

changes in the state of vegetation in subsequent years. Creating maps of various indicators made it possible to identify the factors that influenced the dynamics of changes between 2017–2021 and 2022–2023. Vegetation has become the main determinant of the intensified drought phenomenon that occurred in the last few years. The poor hydrographic network and the presence of few and small water reservoirs were best visualized in the case of the NDWI indicator in variant II. The lowering of the groundwater level, as a result of many years of drainage from the zinc and lead ore mines in Olkusz, proves the occurrence of hydrogeological drought in all years covered by the analysis. However, the number of water bodies increases slightly in the western part of the study area, which may be related to the location of the area beyond the influence of the main depression cone. Until the beginning of 2022, mine waters were discharged into nearby surface watercourses, e.g., to the Baba River and through it to the Sztoła River. The conditions prevailing in this period in the study area did not significantly limit the development of vegetation and did not cause its clear degradation.

The application of remote sensing methods to monitoring drought within the region proved to be invaluable, echoing the sentiments expressed by Pettorelli et al. [51], who emphasized the potential of satellite remote sensing in large-scale ecological assessments. The advantage of making large-scale and long-term analyses of the state of the natural environment via optical imaging is indisputable. These methods, through which individual indicators were calculated, have enabled the estimation of changes in the state of vegetation over the years. Ji and Peters [52] demonstrated a similar application in the Northern Great Plains, where vegetation and drought indices provided crucial insights into vegetation responses over different timeframes. Our approach in applying these indicators, particularly the NDWI, as discussed by [31], clearly highlighted the region's poor hydrographic network and scarce water reservoirs. The NDWI's capacity to monitor vegetation liquid water was indispensable in visualizing these features, especially in relation to hydrogeological drought exacerbated by many years of drainage from the zinc and lead ore mines in Olkusz. The insights from [53] on global greenness trends in semiarid areas further contextualize the observations of this study. Their findings underscore the resilience and vulnerability of vegetation in the face of varying hydrological conditions. In our study area, we observed that, despite the impact of these long-standing drought conditions, vegetation degradation was not significantly apparent until 2022. This observation was even more pronounced with the continuous discharge of mine waters into nearby surface watercourses, such as the Baba River and, subsequently, the Sztoła River. Furthermore, the intensification of the global water cycle, as pointed out by Huntington [54], brings to light the long-term implications of human interventions like mine drainage. While the western part of the study area showed a slight increase in water bodies, possibly because of its location beyond the influence of the main depression cone, the larger region reflects a more intricate relationship between human activity and natural hydrology.

In summary, remote sensing, as both our findings and the established literature suggest, offers a holistic perspective on the dynamics of vegetation and water resources. Through the comprehensive lens of satellite indices and on-the-ground observations, we can better understand and predict the evolving landscape and its implications for the future.

Analyzing all the determined indicators, it can be concluded that the results obtained for 2022 differ significantly from the results obtained for previous years. On this basis, it can be concluded that the cause of the situation was the disconnection of the dewatering pumps of the Pomorzany Mine at the turn of 2021 and 2022. Calculating the NMDI index enabled the assessment of soil moisture over the analyzed years. High soil hydration in 2017–2021, in contrast with its significant drying in 2022, may indicate an agricultural drought at that time. The lack of appropriate conditions for plant growth and development could have been the cause of a significant decline in the condition of the vegetation in the year in question. This is confirmed by the results obtained for vegetation and water indices—on their basis, deterioration in the health of the plant cover and a decrease in the water content in plants can be observed. In addition, the analysis of the state of vegetation in the vicinity

of the Sztoła River Valley in 2017–2022 proves the occurrence of unfavorable changes in the environment. Significant differences in the condition of the vegetation cover within Sztoła, compared with Biała Przemsza and other forest areas, may be a sign of hydrological drought. An additional factor affecting the degradation of the natural environment in 2022 could also be weather conditions. The decline in the condition of the vegetation in the discussed year could be partly related to the extremely dry May and the relatively low rainfall in April and June. The lowest rainfall, combined with the very high temperatures of May and June 2022, most likely caused the drought to deepen. Meteorological drought could delay the start of the growing season, which is confirmed by the illustrations made for MSAVI2. Visible changes in vegetation cover—low vegetation interspersed with bare soil—may indicate the start of plant growth and development in later months. However, the results for indices from May 2023 confirm the observations from 2022 despite high precipitation during that month.

The complicated geological structure, the almost complete drying of the Quaternary stage as a result of the drainage of the Olkusz mines, the presence of a depression funnel, the lack of new mine water supplies due to the disconnection of the drainage system of the Pomorzany Mine [55], and the weather conditions that occurred in 2022 led to significant changes in the entire ecosystem of the region. The presented factors caused the occurrence of four types of drought within the analyzed area: hydrogeological drought, with the longest time range, resulting from lowering the groundwater level; agricultural drought, manifested by insufficient soil moisture, preventing the provision of appropriate conditions for plant development; hydrological drought, characterized by reduced water flow in rivers or their drying up; and meteorological drought, resulting from a significant decrease in precipitation and high temperatures, in particular in May 2022. The combination of both factors—human and natural—caused the occurrence of unfavorable changes in the natural environment of the Olkusz region, the effect of which was visible on a large scale. The results from the year 2023 suggest the low significance of the impact of weather conditions on the intensification of the drought phenomenon.

To mitigate the impact of drought in the Olkusz region, the adoption of an integrated water resource management approach is imperative, emphasizing both conservation and optimized utilization. This encompasses the enhancement of agronomic practices through the introduction of drought-resistant crop varieties and the implementation of water-efficient irrigation systems. Additionally, the promotion of rainwater-harvesting techniques and the rehabilitation of local aquatic ecosystems are crucial for re-establishing hydrological equilibrium. The rehabilitation of local aquatic ecosystems is another critical element in this integrated approach. Restoring wetlands, rivers, and lakes enhances their capacity to store water, thereby improving the region's hydrological balance. These ecosystems also play a vital role in maintaining biodiversity and providing natural filtration for water quality improvement. Efforts to rehabilitate these ecosystems should be coupled with measures to prevent pollution and the over-extraction of water resources.

The experiences and findings from the Olkusz region's drought phenomena offer a strategic framework for other mining-dominated regions. The execution of proactive water management strategies, including the meticulous monitoring and regulation of mine water discharge and the reinforcement of adjacent ecological systems, is vital. The integration of remote sensing methodologies for the precocious detection of environmental alterations is a critical component in facilitating anticipatory measures against drought risks. These methodologies, in conjunction with concerted endeavors among mining corporations, community entities, and environmental regulatory bodies, can substantially diminish the probability of analogous drought occurrences in other mining-intensive areas.

As a consequence of, among other factors, the 2030 Agenda for Sustainable Development [56] and the European Union's response to this document [57], Poland is undergoing a partially compelled mining sector/energy transformation. It affects the number of mines that are already closed or will close in the near future, meaning that the number of areas that need to be reclaimed and redeveloped will only rise. The largest coal-based region

in the EU—Silesia—is in southern Poland [58], with 18 coal mines operating in 2017 [59]. Furthermore, Skoczkowski et al. [59] indicate that this region will encounter challenges associated with adapting to climate change. This is a region with high anthropogenic pressure, for example, the changes that have been observed in the Wojkowice region, where extensive mining activities have led to significant alterations in groundwater dynamics. The groundwater table has decreased by over 60 m, accompanied by a deterioration in its chemical composition, necessitating water purification efforts [60]. Another example highlighting the need for environmental monitoring in the region pertains to areas characterized by acid mine drainage, as exemplified by the case of the Wiśniówka mining area (southcentral Poland) [61]. In Poland, there are also open-pit coal mines, where dewatering practices are employed, as exemplified by the case of Turow (southwest Poland). The activities of this mine have become the subject of a political conflict involving the European Union. One of the accusations pertains to the impact of the mine's operation on the water table and the formation of a depression cone [62].

The utilization of remote sensing is a crucial component of the environmental impact monitoring of mining activities, as well as assessing the effects of revitalization and remediation efforts in postmining areas. This is corroborated by the abundance of research published in these domains, particularly with respect to witnessing an increase in the period between 2010 and 2019, with the NDVI being the most utilized index [63]. This is further corroborated by studies conducted in regions proximate to the area analyzed in this study, such as research pertaining to the former Babina Mine in western Poland [64] or the active Bełchatów open-pit mine [65]. Significant impacts of mining activities on vegetation cover were noted in these studies.

While the application of remote sensing methods in drought research is not novel, the depth and breadth of our analysis present several innovative contributions. To start with, our study uniquely provides a multifaceted examination of drought in the Olkusz region, encompassing hydrogeological, agricultural, hydrological, and meteorological droughts. This comprehensive perspective offers a more in-depth understanding of the interactions between different drought phenomena in a single region. Moreover, the emphasis on the effects of mining activities, particularly their role in inducing hydrogeological droughts, is a distinctive angle that has not been frequently explored. By combining this human-induced factor with natural drought influencers, we present a balanced and encompassing view of the drought sources in the area.

## 6. Conclusions

Many years of exploitation of zinc and lead ore deposits in the Olkusz region led to the lowering of the groundwater level, which, in turn, was associated with the appearance of an extensive depression crater and the significant transformation of the landscape. However, the most severe changes in the natural environment occurred after the closure of the last of the mines discussed—the Pomorzany Mine. At that time, the pumps dewatering the aforementioned mine were disconnected; the drained water had supplied the nearby surface watercourses, e.g., the Sztoła River. The shutdown of the mine drainage system at the beginning of 2022 caused the drying up of this and other rivers and a significant reduction in the flow of the others, which ultimately led to the disruption of the entire ecosystem of the research area. The use of satellite images of the Sentinel-2 mission made it possible to show the disturbing changes that have recently taken place in the Olkusz region. The use of various remote sensing indices—vegetation (NDVI, MSAVI2), water (MSI, NDWI in two variants), and soil (NMDI)—allowed us to conclude that all types of drought occurred within the analyzed area. Poorly hydrated soil (agricultural drought) associated with the presence of a depression cone (hydrogeological drought) and the lack of mine water supply (hydrological drought), combined with unfavorable weather conditions (meteorological drought), also led to the significant deterioration of the vegetation. Expanding the area of research would make it possible to compare the condition of the natural environment in areas not affected by the occurrence of a depression

cone. In addition, the extension of the Olkusz region to the western areas would allow us to estimate the extent to which the flow of the Biała Przemsza River has decreased and to present the related effects.

**Supplementary Materials:** The following supporting information can be downloaded at: https://www. mdpi.com/article/10.3390/rs16010111/s1, Figure S1: Full set of NMDI Soil maps used for analysis. Figure S2: Full set of MSAVI2 maps used for analysis. Figure S3: Full set of MSI maps used for analysis.

**Author Contributions:** Conceptualization, M.L.; methodology, M.L. and A.P.; validation, J.S., M.L. and K.A.; formal analysis, M.L., J.S. and A.P.; investigation, A.P.; resources, A.P.; data curation, A.P. and J.S.; writing—original draft preparation, A.P., M.L, M.M., J.S. and K.A.; writing—review and editing, M.L., M.M., J.S. and K.A.; visualization, A.P. and J.S.; supervision, M.L. and M.M.; project administration, M.L.; funding acquisition, M.M. All authors have read and agreed to the published version of the manuscript.

**Funding:** This research project was partly supported by program "Excellence initiative—research university" for the AGH University of Krakow. This work was also financed within the framework of the statutory research of the AGH University of Krakow, Faculty of Geology, Geophysics, and Environmental Protection.

**Data Availability Statement:** The datasets utilized and examined in the present study can be obtained from the corresponding authors upon reasonable request. The full dataset is not publicly available due to its' volume.

**Conflicts of Interest:** The authors declare no conflicts of interest.

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
