# Peer review of "Why Rivers Disappear—Remote Sensing Analysis of Postmining Factors Using the Example of the Sztoła River, Poland"

_remotesensing, doi:10.3390/rs16010111_

Round 1

Reviewer 1 Report

Comments and Suggestions for Authors

This study provides a comprehensive analysis of drought conditions in the Olkusz region, emphasizing the significant impact of both human-induced factors, particularly mining activities, and natural phenomena on the area's ecosystem. The use of remote sensing techniques, specifically satellite imagery from the Sentinel-2 mission, in conjunction with various indices, offers a detailed examination of different drought types, including hydrogeological, agricultural, hydrological, and meteorological droughts. The study's multidimensional approach to analyzing various types of drought phenomena in the Olkusz region is commendable. The incorporation of hydrogeological, agricultural, hydrological, and meteorological drought aspects provides a holistic understanding of the region's challenges. However, despite the authors' efforts, this study's applicability to other regions is limited. Additionally, the images in the article are blurry and not clear, making it challenging to interpret the visual data. Here are some suggestions for the authors consideration, which I hope will be helpful for the research:

1. Improve Image Quality: please provide clear, high-resolution images. This will help readers better understand the charts and visuals used in the study.

2. Provide Broader Applications: In the discussion section, consider comparing the study results with situations in other regions to illustrate the generalizability of the research.

3. Including a section on potential mitigation strategies or adaptive measures that could be employed to alleviate the impact of drought in the Olkusz region would enhance the practical implications of the study.

Minor Comments:

1. The caption for Figure 2 appears to be too simplistic. It is advisable to provide detailed explanations for the abbreviations used in the figure.

2. There are still red fonts in Table 1; it seems this is not a final version.

3. Lines 238-239 and Lines 244-246 consist of a single sentence each, occupying separate paragraphs. It is suggested that the author reorganize the structure.

4. The format of the references is not consistent; for example, some references have all words in the titles capitalized, while others only capitalize the first word. It is advised that the author carefully check and modify the references for consistency.

Author Response

Dear Reviewer, 

Thank you very much for your valuable feedback on our manuscript. Your insightful comments were greatly appreciated and have helped us improve the quality of our paper. 

We have diligently addressed each of your points to ensure our revised manuscript aligns more closely with your expectations. It is our hope that these changes meet your approval and that the paper is now suitable for publication. 

We are grateful for the time and effort you took to review our work and for guiding us in enhancing its quality. 

Please refer to the point-by-point answers attached. 

Reviewer 2 Report

Comments and Suggestions for Authors

The paper aims to monitor the agricultural, hydrogeological, meteorological, drought in the post-mining areas of Olkusz region, using remote sensing indices. The results are promising, concerning the in-depth understanding of the effects of mining activities.

However, the methodology needs improvement. How did you validate the results? A methodological flowchart must clearly explain all the steps.

All in all, my recommendation is to accept the paper for publication, subject to major revision. Please find in the attached file the amendments which I believe are required prior to accepting the paper.

General comments

-          All figures must be qualitatively improved (very low resolution. The comments on figures can't be seen).

Author Response

Dear Reviewer,
Thanks a lot for your feedback on our paper. Your comments were really helpful, and we've made the changes you suggested to improve it.
We hope these tweaks make the paper better and up to your standards now. We appreciate the time you spent reviewing our work and guiding us on how to make it better.
Check out the detailed responses in the attached document.

Reviewer 3 Report

Comments and Suggestions for Authors

This paper presents a comprehensive analysis of drought in a former mining region in Poland using a variety of remote sensing data sources. The methods are sound and the process is explained well. I recommend publication after the authors correct a few issues. I have marked up the PDF copy of the manuscipt with a number of comments that I believe will improve the readability of the paper. One significant problem is that all of the figure had such a poor resolution that I could not read any of the text on the figures. This could be a result of how the PDF was prepared and should be easy to fix. 

Comments on the Quality of English Language

Overall, the English was good, but I did make quite a few suggestions to improve the English in the annotated PDF. 

Author Response

Dear Reviewer, 

Thank you for your feedback on our manuscript. Your comments were very helpful, and we've made the changes accordingly to elevate the quality of the paper. 

We've taken each of your suggestions seriously to get the manuscript more in line with what you're looking for. We really hope these adjustments hit the mark and make the paper publication-ready. 

We're genuinely grateful for your time and effort in reviewing our work and providing us with valuable insights. 

For a detailed rundown of the changes, please refer to the attached responses. 

Round 2

Reviewer 1 Report

Comments and Suggestions for Authors

I am pleased to observe the authors' responses almost three months later. I express my gratitude to the authors for generously sharing their research findings, and for their thorough addressing of my previous inquiries. I do not have any additional suggestions for modification.

Reviewer 2 Report

Comments and Suggestions for Authors

Your corrections have my full agreement